# Gender, Shame, and Social Support in LGBTQI+ Exposed to Discrimination: A Model for Understanding the Impact on Mental Health

**Joana Cabral ***  and **Tiago Miguel Pinto**

Digital Human-Environment Interaction Lab–HEI-Lab, Lusófona University, 1749-024 Lisboa, Portugal
* Correspondence: joana.cabral@ulusofona.pt

**Abstract:** Discrimination and homonegativity have been consistently linked to poorer mental health outcomes in LGBTQI+ individuals. However, little is known about the role of internal shame and the potential moderating role of social support. This cross-sectional study investigated the impact of discrimination, internal shame, and social support on mental health outcomes in LGBTQI+ individuals, exploring the intersection between gender and sexual orientation. LGBTQI+ participants, especially women, reveal higher levels of discrimination and shame and a stronger impact on mental health outcomes compared to heterosexual counter-partners. Internal shame was found to mediate the impact of discrimination on depression and anxiety. Social support was found to buffer the impact of discrimination on internal shame, depression, and anxiety. These findings have important implications for clinical practice with LGBTQI+ individuals, suggesting that addressing internal shame and building social support networks are central to promoting resilience and mental health. Results also highlight that gender and sexual orientation should be considered in an intersectional approach when addressing gender-based violence and discrimination and its impact on mental health.

**Keywords:** gender-based violence; discrimination; mental health; LGBTQI+; shame; social support

## 1. Introduction

Both gender and sexuality are central to Gender-based violence (GBV), which disproportionately affects women and those whose gender identity or expression does not align with heterosexist norms (Haynes and DeShong 2017; West 2013). The rights of people throughout the LGBTQI+ umbrella are frequently violated in many societies worldwide, exposing this population to daily experiences of discrimination and inequality (Hubbard 2020; Walters et al. 2020). For example, in Portugal, despite the growing acceptance of non-heterosexuality and recent political and legislative changes, LGBTQI+ people continue to face various forms of interpersonal and institutional discrimination. LGBTQI+ discrimination can be experienced in several areas: school, social relationships, workplace, and health services (ILGA 2020; Gato et al. 2021). Evidence shows that most LGBTQI+ youth report experiences of verbal harassment at school, and some describe being physically assaulted (e.g., punched, kicked, or wounded with a weapon (Pizmony-Levy and Kosciw 2016). In addition, many LGBTQI+ youths experience rejection from parents, friends, and peers (Hall 2018). Moreover, several studies reveal that LGBTQI+ people are often the target of professional discrimination and unequal treatment in the recruitment and selection process (Ozeren 2014; Everly et al. 2016). For example, studies found that women with LGBTQI+ identification on their CV were discriminated against, receiving 30% fewer return calls than other women (Luiggi-Hernández et al. 2015), and that LGBTQI+ people report being the target of "jokes" and sexual harassment during recruitment and selection (Mishel 2016). LGBTQI+ individuals are also exposed to prejudice in health services, which can decrease seeking help and adhering to treatment, impacting their health (Dahlhamer et al. 2016).

While most research focuses on gender-based violence (GBV) and discrimination against women and the LGBTQI+ population, there is a lack of studies exploring the phenomena from an intersectional perspective. Such a perspective is crucial in understanding how gender and sexual orientation intersect to create specific vulnerabilities or protective conditions. Which is often overlooked when assessing mental health impacts, resilience, and access to protective factors. This omission obscures our understanding of the unique intersections between gender and sexuality and their role in susceptibility to discrimination. It further obscures how the mental health of distinct groups is affected, the mechanisms involved, and how they can access specific resilience resources. Our study aims to bridge this gap by investigating the differential experiences of groups with various intersections of gender and sexual orientation, focusing on exposure to discrimination, internalized shame, depression, anxiety, and the protective effects of social support. Specifically, we seek to understand how GBV and other forms of targeted violence impact LGBTQI+ persons and how to provide culturally competent resources for help and safety for all survivors.

The cultural organization of strict categories of masculinity and femininity is a regulating device of social roles within patriarchal societies. As a hegemonic system, patriarchy rules most intimacy and sexuality interactions and individual and collective identities (Gilligan and Snider 2018). Within this system of beliefs, sexuality, intimacy, and families assist a simplified version of the evolutive needs of procreation. These essentialist views of sexuality, sexes, and genders in which gender is understood as nature-given also lead to a binary understanding of sex-gender as a dichotomy of strictly male or female bodies and biologies and categorizes people on one of two discrete, polarized, and disconnected classifications of masculine and feminine. This binary framework similarly applies not only to the biological sexes and gender roles but also to sexuality, affect, and intimacy, imposing a heterosexist norm and morality that prescribes opposite sexes and genders' interactions and organizations. Heterosexism has, therefore, historically caused and reproduced a wide range of stereotypes, prescribing the acceptable identities and expected behaviors and marginalizing any other type of nonconforming expressions and existences (Marchia and Sommer 2019). Most who do not conform to a binary gender and hetero and mono-normative sexualities and intimacies are exposed to discrimination and other forms of unequal treatment (Ferrari et al. 2021).

Feminist approaches to GBV have examined how relations of power account for women's increased vulnerability. However, many reproduce heteronormativity, focusing primarily on intimate partner violence or sexual assault and overlooking trans, queer, and non-binary victims, failing to account for the multiple forms in which gender and sexuality are implicated in GBV (Haynes and DeShong 2017). A still sparse but increasing body of research has, however, started to explore variations in the experiences of discrimination and violence among LGBTQI+ subgroups, including those attributed to gender differences. Some evidence points to pervasive heterosexist, gendered, and essentialist motifs and aggressions. Results from a systematic review (Rothman et al. 2011) reveal a higher prevalence of childhood and adult sexual assault victimization for lesbian or bisexual women, while men reported higher hate crime-related sexual assault. Additionally, despite inconsistencies, research points to variations in exposure to discrimination among monosexual and bisexual LGBTQI+ individuals (e.g., Bostwick et al. 2014). Evidence also reveals that transgender women, particularly those of color, experience even more disproportionately high levels of discrimination, suggesting an intersection between racist, heterosexist, and transphobic forms of oppression (e.g., Smart et al. 2022).

The association between LGBTQI+ gender and sexual identities and mental health outcomes has been supported extensively and consistently (Sandfort et al. 2014; Williams et al. 2021). As with exposure to GVB and discrimination, common clinical mental health symptoms among the LGBTQI+ communities seem to vary depending on gender and sexual identity. Research on the impact of discrimination on mental health in LGBTQI+ communities demonstrates that LGBTQI+ individuals have higher rates of chronic illnesses, clinical mental health symptoms, namely depression and anxiety (Han et al. 2020;

Lozano-Verduzco et al. 2017), higher rates of suicide (Fontanella et al. 2015), risky sexual behaviors (Ballard et al. 2017), and substance abuse (Watson et al. 2019a). Moreover, despite inconsistencies, evidence suggests mental health outcomes might vary for sexual minorities (Bostwick et al. 2014). Adding to this, the prevalence of mental health disorders among LGBTQI+ communities also seems to vary as a function of access to social support (Henry et al. 2021; Watson et al. 2019a) and living in a rural or urban environment (Ballard et al. 2017). Again, such disparities underscore that discriminatory experiences and impacts demand a comprehensive and intersectional inspection that considers sexual orientation and gender, race, and class and access to resilience recourses.

According to The Minority Stress Theory (Meyer 2013), mental health problems among LGBTQI+ may be explained by an accumulation of stressors that goes beyond those typical (e.g., loss of a family member, illness, loss of a job) and that includes stressors specific to their minority of nonconforming sexual identity. These specific stressors include (i) situations of discrimination per si (e.g., harassment, violence, discrimination); (ii) anticipation of discrimination and rejection; (iii) pressure to omit identity; (iv) internalization of society's negative attitudes and beliefs (e.g., internalized homonegativity).

Several studies have consistently pointed out that the experiences of being marginalized, isolated, excluded, and bullied create significant social stress for LGBTQI+ people (Hafeez et al. 2017; Schmitz et al. 2020; Felner et al. 2020). The experience of structural and institutional discrimination (e.g., school, work, health, and social services) poses significant psychological challenges, resulting in internalized feelings of inferiority or trans, bi, or homonegativity (Russell and Fish 2016). Faced with hostile environments, many LGBTQI+ decide to conceal their identities to prevent the experiences of rejection, harassment, and discrimination (Herek and Garnets 2007). Concealing identity implies continuously monitoring others' responses and relationships, anticipating safe and unsafe environments, and considering the positive and negative aspects of identity revealing or concealing. These processes of ongoing monitorization, identity concealing, and invisibility require considerable cognitive and emotional effort, therefore, overburdening LGBTQI+ well-being (Herek and Garnets 2007). In a study by Oginni et al. (2018), internalized homonegativity and perceived stigma were associated with depression in homosexual students, accounting for an additional 14% in the variance of depression.

As with many other forms of cultural and identity-related violence resulting from hegemonic and normative pressures, LGBTQI+ discrimination may result in an internalization of sociocultural prejudice, predisposing LGBTQI+ people to perceive, even if non-consciously, their identity and desires as shameful, abnormal, immoral, or a symptom of a mental disorder. These LGBTQI+ negativity internalizations work as an internalized and self-directed form of oppression and have been associated with depression and anxiety (Herek et al. 2015; Newcomb and Mustanski 2010). In addition, society and internal stigma and shame can create concrete and psychological barriers that prevent LGBTQI+ access to mental health services, as evidence shows embarrassment and fear of stigmatization are among the reasons behind young people from sexual and gender minorities' unwillingness to seek support from mental health services (McDermott et al. 2015; Brown et al. 2016).

Internalized homophobia dynamics may coexist with more pervasive feelings of internal shame, amplifying feelings of inadequacy that extend beyond sexual orientation to the essence of self-value. Internal shame represents a deep-seated, crippling self-perception of unworthiness, often emerging from early interactions with caregivers and not confined solely to sexual stigma or shaming (Gilbert 2022). As a self-conscious emotion, shame is embedded in emotional socialization and strongly influenced by these primary relationships. To various degrees, caregivers' socialization of shame may reflect prevailing social and cultural norms, with expected variations on how social stigma is early imprinted (Tangney et al. 2007). These foundational experiences build the template for subsequent shame experiences, calibrating the sources and triggers of shame in response to external cues. Simultaneously, these early caregiving experiences structure emotion regulation strategies, including those required to manage feelings of shame (Gross 2015). Given the profound and

overarching impact such experiences have on personality and identity development, they also establish a varying degree of resilience or vulnerability to discrimination and prejudice. Hence, understanding the intricate intersection of internal shame and homonegativity can shed light on the complex interplay of risk and protective factors when confronting discrimination, including sexual and gender prejudice, and managing its repercussions on mental health (Luthar et al. 2015).

While evidence suggests that LGBTQI+ people are significantly impacted by the discrimination they experience, it also indicates that some protective factors (individual, relational, and community-based) can help to buffer the physical and mental health consequences of these experiences and promote their well-being (Johns et al. 2018). Among the individual protective factors is identity resilience, a stable self-schema that combines a positive appraisal of self, a sense of cohesion, continuity, and self-efficacy. Identity resilience, as identity itself, depends on the interplay between personal and social representations and experiences (Breakwell 2020). It includes a subjective and internal representation of self-worth and value, the willingness and ability to maintain identity despite changes, and a positive self-construal of the self as distinctive from others. Consequently, identity resilience might be established on low levels of internal shame, offering a shield against the negative impacts of sexual and gender discrimination and internalized homophobia. In a study with gay men, identity resilience was negatively associated with internalized homonegativity (Breakwell and Jaspal 2022). This same study also shows that identity resilience and internalized homonegativity are negatively impacted by perceived social discrimination and positively by social support. Furthermore, evidence on the potential mechanisms underlying the relationship between discrimination and mental health among lesbians and gay men points to a trend of gender variabilities in men and women's internalized homonegativity and rejection, which can contribute to the observed disparities (Feinstein et al. 2012).

Relational protective factors, such as those related to family and friends, also play a significant role. Adolescents' nonconforming gender identity or sexual orientation is often a substantial stressor at the family level (Newcomb et al. 2019). A resilient family is often associated with greater support and better mental health for LGBTQI+ youth. On the contrary, the lack of support from the family is strongly associated with mental health problems, suicide, substance use, and sexual risk behaviors (Ryan et al. 2010). Parental and family cohesion and support are associated with higher self-esteem and healthier sexual experiences (Stotzer et al. 2014), better mental health (Veale et al. 2017), and less substance use (Watson et al. 2019b) in LGBTQI+ youth. In a study by Veale et al. (2017), transgender youth between the ages of 16 and 24 who had family support reported lower rates of depressive symptoms and suicide attempts than those whose families were unsupportive.

In addition to the family, studies demonstrate that support from friends and at a community level is crucial and associated with lower rates of potentially health-damaging behaviors (Watson et al. 2019b). Peer support is, in fact, one of the most relevant protective factors to the mental health of LGBTQI+ people who lack family support (Parra et al. 2018). In a Canadian survey on the health of transgender youth, 79% of transgender youth reported choosing a friend when needing help and advice, and 84% of youth reported that their friends helped provide support (Veale et al. 2015). Similarly, integration at school was associated with better mental health among transgender youth (Veale et al. 2015). Youths with greater attachment to school reported good or excellent mental health compared to those with weaker feelings of attachment to school. Furthermore, having a supportive relationship with an educator was associated with lower school absenteeism in transgender youth (Greytak et al. 2013) and with greater feelings of safety when at school (McGuire et al. 2010). Evidence also points to variation in mental health depending on living in a rural or urban environment which might be related to access to and quality of social support (Ballard et al. 2017). These findings, hence, underscore that the prevalence of mental health disorders among LGBTQI+ communities varies not only as a function of access to social support (Henry et al. 2021; Watson et al. 2019a) but as a function of systemic and contextual

opportunities for socialization within safe communities and spaces that include other LGBTQI+ peers and LGBTQI+ trained professionals. Adding to these, stress responses, including seeking social support, may vary between genders. A "tend-and-befriend" response is potentially more common in women, while men may exhibit more of a "fight-or-flight" response, with LGBTQ women being more likely to seek and use social support, while men exhibiting a more confrontational or isolationist approach to stressors may be less prone to seek others as sources of support (Taylor 2012). These gendered coping styles, which are also a consequence of heterosexist and patriarchic socialization, could further impact how LGBTQI+ men and women respond to and are affected by discrimination.

Taken together, findings suggest that an experience of internalized shame may explain the impact of discrimination on LGBTQI+ mental health. In addition, social support may have a protective effect, buffering the impact of exposure to discrimination on mental health. Few studies explored the association between an internalized experience of shame and mental health in LGBTQI+ people (e.g., Matos et al. 2017) since most studies have focused on self-esteem and internalized homonegativity. While self-esteem and internalized homonegativity may function at a more readily and conscious level, internal shame resides at a more core, transversal, and less consciously mentalized level and, hence, may have an enduring impact on the affect directed to the self that bypasses more conscious and rational appreciations of the right to outness, to a non-confirmative identity and of personal value. Moreover, internalized homophobia and internal shame may coexist, intensifying each other. Internal shame may amplify the impact of discrimination and potentiate stigma internalization. Internalized homophobia could, in turn, foster a more profound, generalized sense of unworthiness and shamefulness. These confluence and synergetic dynamics may intensify the deleterious effects of discrimination on mental health.

While much evidence establishes the role of internalized homophobia in the impact of discrimination on mental health, research on the role of internal shame is almost non-existent. Similarly, studies exploring the protective effect of social support on discrimination considering internal shame are also absent from the literature. In addition, most research focuses on gay men and does not explore differences in discrimination and vulnerability that may derive from intersections between gender and sexual orientation. To fill these gaps, our study aims to explore further the negative impact of exposure to discrimination on mental health, and specifically: (1) if LGBTQI+ people are at increased risk when exposed to discrimination, considering mental health, i.e., impacts on anxiety, depression, and internal shame, and if these risks vary as a function of gender; additionally, we explore (2) the mediating role of internal shame in the impact of exposure to discrimination on mental health; and (3) the moderating role of social support in the impact of exposure to discrimination on mental health.

## 2. Materials and Methods

### 2.1. Participants and Procedures

The sample comprised 114 participants, 48.2% were LGBTQI+, and 62.4% were female. LGBTQI+ was coded for participants who self-identify as cis-gender or non-binary (e.g., non-binary, queer, trans, fluid) or with any non-heterosexual orientation (e.g., lesbian, gay, homosexual, fluid, bisexual, asexual).

Participants were recruited through digital social networks and snowballing procedures. Data were collected using an online platform (Google Forms), made available between May 2019 and August 2021. Consent forms were complied with by all participants, who were informed about the study's aims and the voluntary, confidential, and private terms of the participation. The option "I prefer not to answer" was available for the most sensitive questions. Discrimination was assessed using a measure encompassing various discriminatory experiences beyond those solely attributed to gender and sexuality. This broader approach facilitated the assessment of intersectionality in discrimination experiences by revealing the additional discriminatory burden endured by LGBTQI+ individuals beyond what is experienced due to other conditions (e.g., financial status, appearance).

This measurement strategy also facilitates the soundness and spectrum of the comparative analysis of discrimination between cis-heterosexual and LGBTQI+ individuals and between men and women, offering a more nuanced understanding of the discriminatory phenomena across different identities. Given the study's aims and considering that we intended to explore if the LGBTQI+ and gender (being women) conditions resulted in an increased vulnerability to discrimination, only participants (heterosexual or LGBTQI+) reporting at least one experience of discrimination were included in the study.

*2.2. Measures*

Exposure to discrimination. The Experiences of Discrimination Inventory (IED; adapted from Lisboa et al. 2009 by Antunes et al. 2016) was used to assess exposure to discrimination. It consists of an 18-item self-reported scale assessing the subjective experience of discrimination, alluding to the last year. Participants report contexts (e.g., "Experiencing difficulties and/or discomfort when accessing certain public places (e.g., cafes, bars, museums, theaters).", "Being the target of comments that bother me (e.g., jokes, popular sayings, anecdotes)."), and motives (e.g., "Because of being a man or a woman.", "Because of your sexuality.", "Because of your financial status.", "By your appearance (e.g., weight, height, clothing, ...).") of discrimination. Higher scores refer to a higher sum of discrimination experiences ($\alpha = 0.83$).

Internal Shame. The Internal Shame Scale (ISS; Cook 1996; Portuguese Version, Matos et al. 2012) was used to assess internal shame. It consists of a 24-item self-reported scale scored on a 5-point Likert-type scale referring to internal shame. Higher scores indicate higher levels of internal shame ($\alpha = 0.94$).

Social support. An item scored on a 5-point Likert-type scale assessing social support was extracted from The World Health Organization Quality of Life brief measure (WHOQOL-Bref; European Portuguese version by Canavarro et al. 2007). As the brief version only assesses support from friends, for this study, the item was rephrased to include support from family ("How satisfied are you with the support you get from your friends and family?").

Mental health. The Brief Symptoms Inventory (BSI; Derogatis 1993; Portuguese version by Canavarro 1999) was used to assess mental health. It consists of a 53-item self-reported scale scored on a 5-point Likert-type scale assessing symptoms of psychological distress and psychiatric disorders. For the present study, the 12 items corresponding to depression (e.g., "Feeling sad"; $\alpha = 0.87$) and anxiety (e.g., "Easily getting annoyed or irritated"; $\alpha = 0.62$) were used.

*2.3. Statistical Analysis*

Two-way and one-way ANOVAs were performed to explore (1) the impact of sexual orientation and gender on exposure to discrimination, mental health, shame, and social support. The two-way ANOVA models included sexual orientation, gender, and the interaction between sexual orientation and gender as independent variables (IVs), and exposure to discrimination, mental health, shame, and social support as dependent variables (DVs). Pairwise comparisons were applied using Bonferroni correction to explore significant interaction effects further.

A moderated mediation model was performed using PROCESS SPSS macro (Igartua and Hayes 2021) to explore (2) the mediating role of shame in the impact of exposure to discrimination on mental health and (3) the moderating role of social support on the impact of exposure to discrimination on mental health. À priori power calculations were performed following recommendations (Faul et al. 2007, 2009), revealing the sample size is adequate to conduct the moderated mediation models (f2 = 0.25, $p < 0.05$, N = 104, number of predictors = 9; power = 0.95). The model included exposure to discrimination as IV, shame as the mediator variable, and the interaction between sexual orientation and gender and social support as moderators. All possible interactions between the IV and the

moderators were calculated. Direct and indirect effects of exposure to discrimination on shame and mental health were analyzed.

Statistical analysis was performed using SPSS (Statistical Package for the Social Sciences) version 27.0.

## 3. Results

### 3.1. Participants' Sociodemographic Characteristics

Participants were aged between 18 and 51 years old ($M_{age}$ = 30.31, *SD* = 8.08). Most participants were Portuguese (78.1%) and white (93%) and had more than nine years of schooling (86%). Half (51.8%) identified as cisgender and heterosexual, and the other half (48.2%) as LGBTQI+. More than half were single (55.3%), were from low or medium socioeconomic levels (65.8%) and had a monthly income higher than 1000 € (61.4%; see Table 1). No associations and differences were found between sexual orientation or gender with participants' sociodemographic characteristics.

**Table 1.** Participants' Sociodemographic Characteristics.

|  |  | % (N = 114) |
|---|---|---|
| Sexual orientation | LGBTQI+ | 48.2 |
|  | Heterosexual | 51.8 |
| Gender | Female | 62.4 |
|  | Male | 34.9 |
|  | Trans or non-binary | 2.7 |
| Years of schooling | 6–9 | 13.3 |
|  | 10–12 | 32.5 |
|  | >12 | 53.5 |
| Monthly income | 250 €–500 € | 0.9 |
|  | 501 €–1000 € | 22.8 |
|  | 1001 €–2000 € | 36.8 |
|  | >2000 € | 24.6 |
| Socioeconomic level | Low | 30.7 |
|  | Medium | 35.1 |
|  | High | 0.9 |
| Marital status | Single | 55.3 |
|  | Married/cohabiting/with partner | 40.3 |
|  | Divorced | 4.4 |

Notes. Percentages do not sum to a total of 100% due to missing values.

### 3.2. Independent and Interaction Impact of Sexual Orientation and Gender on Exposure to Discrimination, Mental Health, Shame, and Social Support

Results from two-way ANOVA revealed significant univariate effects of sexual orientation on discrimination, *F*(3,103) = 5.03, *p* < 0.05, η2 = 0.05, and shame, *F*(3,103) = 34.68, *p* < 0.001, η2 = 0.25. LGBTQI+ participants reported higher levels of discrimination and shame than heterosexual participants (see Table 2). No univariate effects were found on depressive and anxiety symptoms and social support. No univariate effects of gender were found on discrimination, depressive and anxiety symptoms, shame, and social support (see Table 2). Significant effects were found on shame for the interaction between gender and sexual orientation (see Table 3). The interaction effects were inspected with a UNINOVA using an interaction variable composed of gender × sexual orientation.

**Table 2.** Independent impact of sexual orientation and gender on discrimination, mental health, shame, and social support.

| | Sexual Orientation | | | | | Gender | | | | | |
| --- | --- | --- | --- | --- | --- | --- | --- | --- | --- | --- | --- |
| | LGBTQI+ (*n* = 48) | | Heterosexual (*n* = 59) | | | Women (*n* = 68) | | Men (*n* = 39) | | | |
| | *M* | *SD* | *M* | *SD* | *F* | *M* | *SD* | *M* | *SD* | *F* | *df* |
| Discrimination | 30.48 | 11.43 | 25.98 | 11.32 | 5.04 * | 27.43 | 10.22 | 29.00 | 13.61 | 0.56 | 3,103 |
| Depressive symptoms | 2.13 | 0.91 | 1.95 | 0.70 | 1.39 | 2.10 | 0.81 | 1.91 | 0.78 | 1.36 | 3,103 |
| Anxiety symptoms | 1.84 | 0.68 | 1.71 | 0.47 | 1.86 | 1.81 | 0.57 | 1.70 | 0.58 | 0.89 | 3,103 |
| Shame | 2.47 | 0.93 | 1.53 | 0.98 | 34.68 *** | 2.01 | 0.93 | 1.85 | 1.27 | 0.53 | 3,103 |
| Social support | 3.67 | 0.93 | 3.70 | 1.15 | 0.00 | 3.81 | 1.01 | 3.46 | 1.10 | 2.55 | 3,103 |

Notes. *M* = Mean; *SD* = Standard deviation. * *p* < 0.05; *** *p* < 0.001.

**Table 3.** The impact of the interaction between sexual orientation and gender on discrimination, mental health, shame, and social support.

| | (1) LGBTQI+ Women (*n* = 30) | | (2) LGBTQI+ Men (*n* = 18) | | (3) Heterosexual Women (*n* = 38) | | (4) Heterosexual Men (*n* = 21) | | | | |
| --- | --- | --- | --- | --- | --- | --- | --- | --- | --- | --- | --- |
| | *M* | *SD* | *M* | *SD* | *M* | *SD* | *M* | *SD* | *F* | *df* | |
| Discrimination | 28.87 | 10.51 | 33.17 | 12.67 | 26.29 | 9.98 | 25.43 | 13.66 | 0.26 | 3,103 | 1 = 2 = 3 = 4 |
| Depressive symptoms | 2.17 | 1.01 | 2.04 | 0.73 | 2.04 | 0.60 | 1.79 | 0.83 | 0.15 | 3,103 | 1 = 2 = 3 = 4 |
| Anxiety symptoms | 1.84 | 0.72 | 1.84 | 0.64 | 1.79 | 0.43 | 1.57 | 0.51 | 0.95 | 3,103 | 1 = 2 = 3 = 4 |
| Shame | 2.32 | 0.96 | 2.74 | 0.85 | 1.78 | 0.84 | 1.08 | 1.06 | 8.98 ** | 3,103 | 1, 2, 3 > 4; 2 < 3 > 4 |
| Social support | 3.77 | 0.93 | 3.50 | 0.93 | 3.84 | 1.08 | 3.42 | 1.26 | 0.13 | 3,103 | 1 = 2 = 3 = 4 |

Notes. *M* = Mean; *SD* = Standard deviation. ** *p* < 0.01.

Results revealed a significant effect of the interaction between sexual orientation and gender on shame, $F(3,103) = 8.98$, $p < 0.01$, η2 = 0.08. LGBTQI+ women and men, and heterosexual women reported more levels of shame than heterosexual men (all *p*s < 0.05). In addition, heterosexual women reported lower levels of shame than LGBTQI+ men and higher levels of shame than heterosexual men (see Table 3). Although no univariate effects of the interaction between sexual orientation and gender were found on discrimination, depressive and anxiety symptoms, shame, and social support (see Table 3), plots with estimated marginal means suggested further interaction effects for these variables (see figures in Supplementary Materials).

*3.3. Impact of Exposure to Discrimination on Mental Health: The Mediating Role of Shame and the Moderating Role of Social Support*

Results from the first step revealed a statistically significant model that explained 59% of the variance of shame, $F(9,95) = 15.19$, $p < 0.001$, $R^2 = 0.59$. Higher exposure to discrimination predicted higher levels of shame, β = 0.25, $p < 0.001$, and higher levels of social support predicted lower levels of shame, β = −0.25, $p < 0.001$ (see Table 4). Participants with more exposure to discrimination and reporting lower social support presented higher levels of shame (see Figure 1). The interaction between exposure to discrimination, sexual orientation and gender, and social support was statistically significant for heterosexual men, with the lowest levels of shame, and LGBTQI+ women, with the highest levels. Results also show a higher impact of discrimination on shame for heterosexual women with low social support. The analysis of Figure 1 additionally reveals that LGBTQI+ men and women report higher levels of shame compared to heterosexual counter-partners and that

heterosexual men and LGBTQI+ women are the most impacted by discrimination. Yet heterosexual men's average levels of shame are the lowest compared to all other participants. Additionally, compared to LGBTQI+ women, LGBTQI+ men who are less exposed to discrimination have higher levels of shame, but LGBTQI+ women who are more exposed to discrimination show similar levels of shame.

**Table 4.** Moderated mediation model results: the mediator role of shame and the moderator role of social support in the impact of exposure to discrimination on depressive and anxiety symptoms.

| | R (R$^2$) | F | β | p | CI 95% |
|---|---|---|---|---|---|
| *Outcome variable—shame* | 0.77 (0.59) | 15.19 *** | | <0.001 | |
| Discrimination | | | 0.25 *** | <0.001 | 0.02–0.08 |
| All (1) vs. heterosexual men (0) | | | 1.48 *** | <0.001 | 0.97–1.98 |
| Women (1) vs. men (0) | | | 0.76 *** | <0.001 | 0.32–1.19 |
| LGBTQI+ women (1) vs. all others (0) | | | 0.43 * | 0.017 | 0.08–0.78 |
| Social support | | | −0.25 ** | 0.001 | −0.40−−0.10 |
| Discrimination × all but heterosexual men | | | −0.04 † | 0.077 | −0.07–0.00 |
| Discrimination × women | | | 0.01 | 0.551 | −0.03–0.05 |
| Discrimination × LGBTQI+ women | | | 0.02 | 0.240 | −0.01–0.05 |
| Discrimination × social support | | | −0.00 | 0.562 | −0.01–0.01 |
| Discrimination × GxSO × social support | | | 1.52 | 0.203 | |
| *Outcome variable—depressive symptoms* | 0.74 (0.54) | 11.07 *** | | <0.001 | |
| Discrimination | | | 0.01 | 0.241 | −0.01–0.04 |
| Shame | | | 0.31 *** | <0.001 | 0.15–0.48 |
| All (1) vs. heterosexual men (0) | | | −0.51 * | 0.035 | 0.04–0.97 |
| Women (1) vs. men (0) | | | 0.48 * | 0.010 | 0.12–0.85 |
| LGBTQI+ women (1) vs. all others (0) | | | −0.11 | 0.441 | −0.40–0.17 |
| Social support | | | −0.04 | 0.493 | −0.17–0.08 |
| Discrimination × all but heterosexual men | | | 0.02 | 0.284 | −0.01–0.05 |
| Discrimination × women | | | −0.02 | 0.205 | −0.05–0.01 |
| Discrimination × LGBTQI+ women | | | 0.04 ** | 0.006 | 0.01–0.07 |
| Discrimination × social support | | | −0.00 | 0.445 | −0.01–0.01 |
| Discrimination × GxSO × social support | | | 2.39 † | 0.056 | |
| *Outcome variable—anxiety symptoms* | 0.65 (0.43) | 7.10 *** | | <0.001 | |
| Discrimination | | | 0.00 | 0.779 | −0.02–0.02 |
| Shame | | | 0.15 ** | 0.002 | 0.02–0.28 |
| All (1) vs. heterosexual men (0) | | | −0.13 | 0.490 | −0.51–0.25 |
| Women (1) vs. men (0) | | | 0.26 † | 0.088 | −0.04–0.55 |
| LGBTQI+ women (1) vs. all others (0) | | | −0.08 | 0.482 | −0.31–0.15 |
| Social support | | | −0.03 | 0.581 | −0.13–0.07 |
| Discrimination × all but heterosexual men | | | 0.03 * | 0.021 | 0.00–0.06 |
| Discrimination × women | | | −0.03 * | 0.016 | −0.05−−0.01 |
| Discrimination × LGBTQI+ women | | | 0.03 ** | 0.006 | 0.01–0.05 |
| Discrimination × social support | | | −0.01 † | 0.083 | −0.01–0.01 |
| Discrimination × GxSO × social support | | 3.73 ** | | 0.007 | |

Notes: GxSO—Gender × sexual orientation. † $p < 0.10$; * $p < 0.05$; ** $p < 0.01$; *** $p < 0.001$.

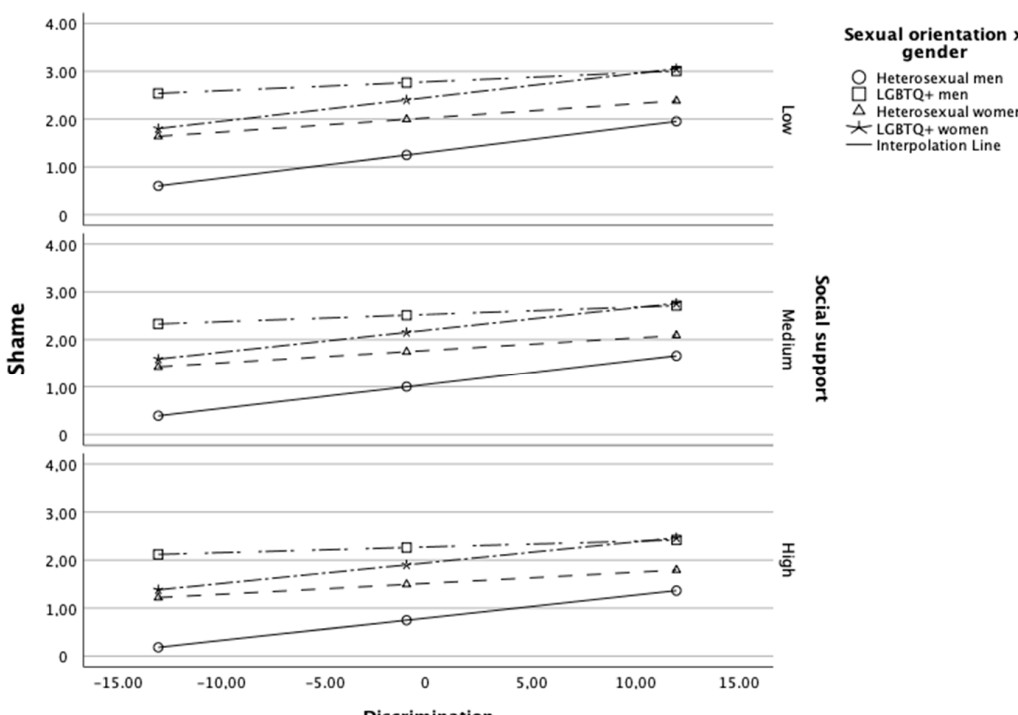

**Figure 1.** The moderator role of social support in the impact of exposure to discrimination on shame.

Results from the second step of the depressive symptoms model (see Table 4) revealed a statistically significant model that explained 54% of the variance of depressive symptoms, $F(9,95) = 11.07$, $p < 0.001$, $R^2 = 0.54$. Higher levels of shame predicted higher levels of depressive symptoms. When interactions between exposure to discrimination and sexual identity and gender were not considered, heterosexual men and heterosexual and LGBTQI+ women showed higher depressive symptoms. The interaction between exposure to discrimination and sexual orientation and gender, significantly predicted depressive symptoms, $\beta = 0.04$, $p = 0.028$. The double interaction between discrimination, gender and sexual orientation, and social support was marginally significant (see Table 4).

Results from the second step of the anxiety symptoms model revealed a statistically significant model that explained 43% of the variance of anxiety symptoms, $F(9,95) = 7.10$, $p < 0.001$, $R^2 = 0.43$. Higher shame levels predicted higher anxiety symptoms for all. The interactions between exposure to discrimination and sexual orientation and gender, and the double interaction between discrimination, sexual identity and gender, and social support significantly predicted anxiety symptoms (see Table 4).

The analysis of conditional effects allows for a better inspection of specific moderated mediation effects of gender and sexual orientation and internal shame in the link between exposure to discrimination and mental health. Conditional direct effects revealed that this interaction was statistically significant for LGBTQ+ women and men (see Table 5). In conditions of low exposure to discrimination, LGBTQI+ men and women revealed lower depressive and anxiety symptoms than their heterosexual counterparts. However, they are more impacted when levels of discrimination are higher (Figure 2A,B), with LGBTQI+ women showing the highest levels of depressive and anxiety symptoms. Conditional effects also show that depressive and anxiety symptoms are lower in conditions of higher social support (see Table 5).

**Table 5.** The impact of exposure to discrimination on shame and depressive and anxiety symptoms: conditional effects by sexual orientation, gender, and social support.

| | Social Support | Effect | p | CI 95% |
|---|---|---|---|---|
| *Internal Shame* | | | | |
| **Sexual orientation × gender** | | | | |
| Heterosexual men (1) | −0.69 | 0.054 *** | 0.0001 | 0.029–079 |
| Heterosexual men (1) | 0.31 | 0.051 ** | 0.001 | 0.021–0.080 |
| Heterosexual men (1) | 1.31 | 0.047 * | 0.013 | 0.010–0.084 |
| LGBTQI+ men (2) | −0.69 | 0.018 | 0.194 | −0.010–0.047 |
| LGBTQI+ men (2) | 0.31 | 0.015 | 0.268 | −0.012–0.043 |
| LGBTQI+ men (2) | 1.31 | 0.012 | 0.443 | −0.019–0.043 |
| Heterosexual women (3) | −0.69 | 0.030 * | 0.023 | 0.004–0.055 |
| Heterosexual women (3) | 0.31 | 0.026 * | 0.033 | 0.002–0.050 |
| Heterosexual women (3) | 1.31 | 0.023 | 0.106 | −0.005–0.051 |
| LGBTQI+ women (4) | −0.69 | 0.050 * | 0.001 | 0.023–0.078 |
| LGBTQI+ women (4) | 0.31 | 0.047 * | 0.001 | 0.021–0.073 |
| LGBTQI+ women (4) | 1.31 | 0.043 * | 0.004 | 0.014–0.073 |
| *Depressive symptoms* | | | | |
| **Sexual orientation × gender** | | | | |
| Heterosexual men (1) | −0.69 | 0.016 | 0.140 | −0.006–0.038 |
| Heterosexual men (1) | 0.31 | 0.013 | 0.306 | −0.012–0.038 |
| Heterosexual men (1) | 1.31 | 0.009 | 0.542 | −0.021–0.040 |
| LGBTQI+ men (2) | −0.69 | 0.034 * | 0.004 | 0.011–0.056 |
| LGBTQI+ men (2) | 0.31 | 0.030 * | 0.008 | 0.008–0.051 |
| LGBTQI+ men (2) | 1.31 | 0.026 * | 0.035 | 0.002–0.051 |
| Heterosexual women (3) | −0.69 | 0.015 | 0.158 | −0.006–0.036 |
| Heterosexual women (3) | 0.31 | 0.011 | 0.259 | −0.008–0.031 |
| Heterosexual women (3) | 1.31 | 0.008 | 0.488 | −0.015–0.030 |
| LGBTQI+ women (4) | −0.69 | 0.053 *** | <0.001 | 0.030–0.076 |
| LGBTQI+ women (4) | 0.31 | 0.050 *** | <0.001 | 0.028–0.071 |
| LGBTQI+ women (4) | 1.31 | 0.046 *** | <0.001 | 0.021–0.070 |
| *Anxiety symptoms* | | | | |
| **Sexual orientation × gender** | | | | |
| Heterosexual men (1) | −0.69 | 0.007 | 0.423 | −0.011–0.025 |
| Heterosexual men (1) | 0.31 | 0.001 | 0.949 | −0.019–0.021 |
| Heterosexual men (1) | 1.31 | −0.006 | 0.636 | −0.030–0.019 |
| LGBTQI+ men (2) | −0.69 | 0.037 *** | <0.001 | 0.019–0.056 |
| LGBTQI+ men (2) | 0.31 | 0.031 ** | 0.001 | 0.013–0.048 |
| LGBTQI+ men (2) | 1.31 | 0.024 * | 0.017 | 0.005–0.044 |
| Heterosexual women (3) | −0.69 | 0.009 | 0.314 | −0.008–0.025 |

**Table 5.** *Cont.*

| | Social Support | *Effect* | *p* | *CI 95%* |
|---|---|---|---|---|
| Heterosexual women (3) | 0.31 | 0.002 | 0.796 | −0.014–0.018 |
| Heterosexual women (3) | 1.31 | −0.004 | 0.628 | −0.023–0.014 |
| LGBTQI+ women (4) | −0.69 | 0.040 *** | <0.001 | 0.021–0.059 |
| LGBTQI+ women (4) | 0.31 | 0.034 *** | <0.001 | 0.016–0.051 |
| LGBTQI+ women (4) | 1.31 | 0.027 ** | 0.008 | 0.007–0.047 |

* p < 0.05, ** p < 0.01, *** p < 0.001.

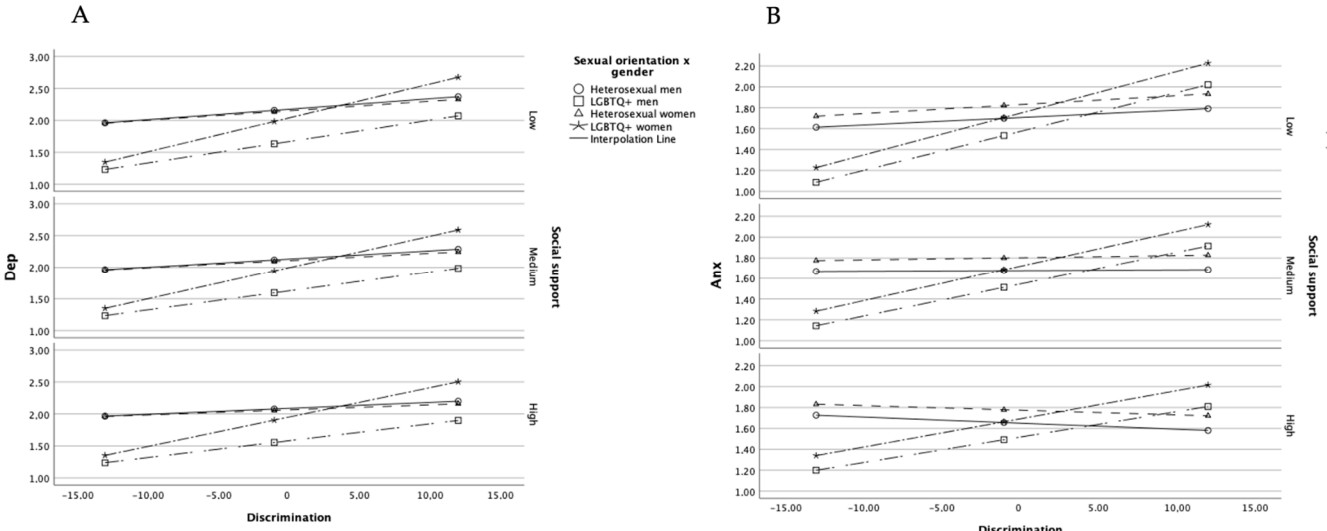

**Figure 2.** The moderator role of social support in the impact of exposure to discrimination on depressive (**A**) and anxiety (**B**) symptoms.

Results from the analyses of the conditional indirect effects of exposure to discrimination on depressive and anxiety symptoms through shame, revealed that the mediator role of shame is only statistically significant for LGBTQ+ women and heterosexual men (see Table 6). These results suggest that while, for heterosexual men and LGBTQI+ women, exposure to discrimination in mental health may be accounted for by its impact on internal shame, it may be independent of internal shame for heterosexual men and LGBTQI+ women.

**Table 6.** The mediator role of shame in the impact of exposure to discrimination on depressive and anxiety symptoms: conditional effects by sexual orientation, gender, and social support.

| | Social Support | *Effect* | *CI 95%* |
|---|---|---|---|
| *Depressive symptoms* | | | |
| **Sexual orientation × gender** | | | |
| Heterosexual men (1) | −0.69 | 0.02 | 0.01–0.03 |
| Heterosexual men (1) | 0.31 | 0.02 | 0.00–0.03 |
| Heterosexual men (1) | 1.31 | 0.01 | 0.00–0.03 |
| LGBTQI+ men (2) | −0.69 | 0.01 | −0.00–0.02 |
| LGBTQI+ men (2) | 0.31 | 0.01 | −0.00–0.02 |
| LGBTQI+ men (2) | 1.31 | 0.00 | −0.01–0.02 |
| Heterosexual women (3) | −0.69 | 0.01 | −0.00–0.02 |

**Table 6.** *Cont.*

|  | **Social Support** | *Effect* | *CI 95%* |
|---|---|---|---|
| Heterosexual women (3) | 0.31 | 0.01 | −0.00–0.02 |
| Heterosexual women (3) | 1.31 | 0.01 | −0.00–0.02 |
| LGBTQI+ women (4) | −0.69 | 0.02 | 0.00–0.03 |
| LGBTQI+ women (4) | 0.31 | 0.01 | 0.00–0.03 |
| LGBTQI+ women (4) | 1.31 | 0.01 | 0.00–0.03 |
| *Anxiety symptoms* | | | |
| **Sexual orientation × gender** | | | |
| Heterosexual men (1) | −0.69 | 0.01 | 0.00–0.02 |
| Heterosexual men (1) | 0.31 | 0.01 | 0.00–0.02 |
| Heterosexual men (1) | 1.31 | 0.01 | 0.00–0.02 |
| LGBTQI+ men (2) | −0.69 | 0.00 | −0.00–0.01 |
| LGBTQI+ men (2) | 0.31 | 0.00 | −0.00–0.01 |
| LGBTQI+ men (2) | 1.31 | 0.00 | −0.00–0.01 |
| Heterosexual women (3) | −0.69 | 0.00 | −0.00–0.01 |
| Heterosexual women (3) | 0.31 | 0.00 | −0.00–0.01 |
| Heterosexual women (3) | 1.31 | 0.00 | −0.00–0.01 |
| LGBTQI+ women (4) | −0.69 | 0.01 | 0.00–0.02 |
| LGBTQI+ women (4) | 0.31 | 0.01 | 0.00–0.02 |
| LGBTQI+ women (4) | 1.31 | 0.01 | 0.00–0.02 |

## 4. Discussion

Despite social and legislative changes aiming at greater protection of the rights of sexual minorities, little improvements are found in the experiences of discrimination that LGBTQI+ people continue to face in various areas of their lives (e.g., Gato et al. 2021). Evidence has consistently shown the impact of discrimination on LGBTQI+ mental health (e.g., Sandfort et al. 2014; Williams et al. 2021). In addition, evidence suggests that an experience of internalized negativity (Timmins et al. 2020; Van Beusekom et al. 2018) may partly explain this impact and that social support may be protective and buffer the impact of exposure to discrimination (Watson et al. 2019a). However, very little research has been conducted on the association between internalized shame and mental health in LGBTQI+ individuals, with most studies focusing on self-esteem and homonegativity. Additionally, no studies were found exploring the protective impact of social support on discrimination considering internal shame.

Internalized shame operates at a non-conscious level and is rooted in early experiences of shame and rejection that occur when children and adolescents express desires and initiatives censured by parents (Matos and Pinto-Gouveia 2014). It may, hence, have an enduring impact on the affect directed to the self that bypasses conscious and rational appreciation of personal value and the right to a non-confirmative identity. In the case of LGBTQI+ children, this can result from early pressures to adhere to heterosexist societal norms vesiculated by parents and society (Rizzuto 2014). The current study tested a model to explore further the negative impact of exposure to discrimination on mental health, proposing internal shame as a central dimension in understanding the effects of exposure to discrimination on the mental health of LGBTQI+ people and exploring the protective role of social support.

Our findings show that LGBTQI+ men and women reported higher levels of discrimination and shame than heterosexual counter-partners. Heterosexual women, while reporting lower shame than LGBTQI+ men, reported more shame than heterosexual men.

No gender and sexual orientation differences were found, independently or when interacting, on depressive and anxiety symptoms and social support. Further analyses, however, testing a moderated mediation, showed specific and significant interaction effects for gender and sexual orientation and that higher exposure to discrimination and lower social support predicted higher levels of shame and depressive anxiety symptoms.

Results showed that shame increases significantly more for heterosexual men and LGBTQI+ women exposed to discrimination. However, while heterosexual men revealed the lowest average levels of shame, LGBTQI+ women showed significantly higher average levels of shame. Results also showed that while LGBTQI+ men's shame is less impacted by discrimination, discrimination affects LGBTQI+ women's internal shame more intensely.

A similar pattern of results was found considering the impact of discrimination on anxiety and depression. Results also revealed that the impact of discrimination on depressive and anxiety symptoms is potentiated in LGBTQI+ people, especially in women, whose depressive and anxiety symptoms are the highest. The results additionally reveal that an increased internalization of shame mediates the impact of discrimination on depression and anxiety for heterosexual men and LGBTQI+ women.

Results also showed that higher social support buffers the impact of exposure to discrimination on depression and anxiety symptoms. Additionally, for heterosexual women in conditions of low and medium social support, internal shame is more impacted by exposure to discrimination.

Our results, showing that the interaction between gender identity and discrimination predicts increased levels of depression and anxiety symptoms, broadly align with previous literature revealing that exposure to discrimination negatively impacts the mental health of LGBTQI+ people (e.g., Henry et al. 2021; Williams et al. 2021). These results also provide evidence for the Minority Stress Theory (Meyer 2013). This theory posits that sexual minorities, including the LGBTQI+ communities, experience several distinct and chronic stressors associated with their stigmatized identities, including victimization, prejudice, and discrimination, in line with previous evidence of minority LGBTQI+ stressors (Sattler et al. 2016). Our sample was composed exclusively of participants who reported experiencing at least one instance of discrimination, regardless of their sexual identity. Among these, our findings indicate a higher prevalence of discrimination and poorer mental health for LGBTIQI+ individuals, highlighting the presence of specific social stressors and discrimination experiences specific to their minority status.

Our results on the mediating role of internal shame contribute to deepening the understanding of the impact of LGBTQI+ discrimination as a pervasive heteronormative violence that persists beyond changes in the legal and formal narrative. In addition to supporting previous evidence on the impact of discrimination on mental health and an increased vulnerability for LGBTQI+ people, our findings also reveal that this impact occurs via internal shame, which may explain the prevalence of the negative effect of internalized homonegativity on LGBTQI+ mental health (e.g., Breakwell and Jaspal 2022; Jaspal et al. 2022). Our results align with previous evidence showing that LGBTQI+ individuals tend to experience more shaming traumatic events and that shame mediates the link between these experiences and poorer mental health (Scheer et al. 2020). Results also align with evidence that gay men recall more shaming experiences with caregivers, especially fathers, and that these insidious and early experiences lead to internal shame and depressive symptoms (Matos and Pinto-Gouveia 2014).

Our findings highlight the importance of considering the intersection of gender and sexual identity when examining mental health outcomes and the effects of discrimination. We found that LGBTQI+ men reported higher shame levels than heterosexual men but were less affected by discrimination. On the other hand, both heterosexual men and LGBTQI+ women were more heavily impacted by discrimination experiences. LGBTQI+ women reveal lower shame than LGBTQI+ men when less exposed to discrimination but are more intensely affected by discrimination. These results align with previous research

showing elevated levels of shame in women (Benetti-McQuoid and Bursik 2005), LGBTQI+ individuals (Scheer et al. 2020), and particularly LGBTQI+ women (Straub et al. 2018).

Additionally, our findings align with studies that show that men who adhere more strongly to traditional gender norms have a higher susceptibility to shame (Gebhard et al. 2019). Possible explanations for these differences may include the socialization of gender roles. Traditional or heteronormative masculinity is closely linked to shame, as status and dominance, on the one hand, and stoicism and invulnerability, on the other, are central expectations for men (Reilly et al. 2014). Men falling out of these traditional roles may be more susceptible to shame (Gebhard et al. 2019). This may lead to a greater tendency towards shame (i.e., internal or trait shame) and greater difficulty regulating these emotions. Additionally, evidence suggests that LGBTQI+ men may experience shame early in their development (Matos et al. 2017), which may explain their higher baseline levels of internal shame.

Despite heterosexual men's privileged social and cultural position, their vulnerability to discrimination experiences is not incompatible with broader data on systemic sexist and homophobic discrimination. Our findings may be better understood given the highly subjective character of the discrimination and shaming experiences. Hegemonic masculinity expectations and pressures (Connell and Messerschmidt 2005) may contribute to exposure or vulnerability to discrimination for heterosexual men who deviate from these norms. Research demonstrates that men's identity concerns often revolve around the threat of violating traditional masculine roles, which can elicit intense feelings of anxiety, shame, and humiliation (Vandello and Bosson 2013). Moreover, male gender role socialization promotes a "shame-phobic" male experience (Reilly et al. 2014), with consequences to mental health, namely internalized shame and depression (Rice et al. 2016), highlighting the complex relationship between shame and adherence to patriarchic masculine norms. Furthermore, some heterosexual men might perceive equality demands and achievements as threatening their privilege or status (Norton and Sommers 2011).

Finally, no gender and sexual orientation differences were found for social support. This is inconsistent with some previous evidence on social support showing that women, both heterosexual and LGBTQI+, are generally more engaged in the community and tend to use support-seeking coping more frequently than men (Pflum et al. 2015). Our findings are, however, following previous literature showing that higher social support predicts better mental health for gays and bisexual men (Henry et al. 2021; Pereira and Silva 2021) and lower levels (external) shame in LGBTQI+ individuals (Seabra et al. 2021). Other studies also indicated that transgender individuals who perceived family support had lower levels of psychological distress than those who perceived their family members as unsympathetic or neutral (James et al. 2016). Similar results were found by Jablonski (2020) when verifying that social support was associated with lower levels of depressive symptoms in LGBTQI+ people. Future research should further investigate the intricate relationship between gender, sexual identity, and social support by adopting an intersectional lens to explore how each condition and identity interact to shape how men and women, heterosexual and LGBTQI+, monosexual and cisgender, and binary and non-binary, receive, and benefit from social support.

Our findings highlight the role of internal shame, mostly overlooked until now, in the minority stress model (Meyer 2013), pointing to its critical role in mental health outcomes when facing discrimination. Furthermore, these findings underline the importance of intersectionality, as they show distinct effects of discrimination, internal shame, and social support across intersections of gender and sexual identity. Our study, hence, adds to the knowledge regarding the minority stress model (Meyer 2013) by revealing internalized shame and social support as a risk and a protective factor, respectively. Furthermore, our results suggest an interaction between these factors, showing that while internalized shame may intensify the impact of discrimination on mental health, social support may buffer this impact by decreasing the effect of internal shame on depression and anxiety symptoms. In sum, results call for an expanded and nuanced understanding of minority

stress that includes both conscious and unconscious experiences of internalized negativity, particularly shame, and the role of social support in buffering these experiences from an intersectional perspective. Differences related to gender and sexual identity should be further explored in future research.

Despite the relevant contributions to the existing literature, our study has limitations that should be mentioned. At first, it fails to fully represent the breadth and diversity of the LGBTQI+ community, as it was limited to specific sociodemographic characteristics such as income and education level. Participants are mostly white with higher education and average income. Thus, findings may not be generalized to other groups within the LGBTQI+ community unrepresented in our sample, especially those that accumulate minority and discrimination-related stressors, such as transgender and non-binary, those racialized, and with low income. Existing literature and evidence highlight increased vulnerabilities at the intersection of gender, race, and sexual minority statuses. Due to these sampling constraints, the severity of discriminatory experiences and their impact on mental health could be underestimated. Moreover, different groups within these intersections may experience unique barriers when seeking social support, specifically those with multiple marginalized identities, such as racialized transgender women. Future research must strive to encompass these diverse experiences following a comprehensive understanding of interceptional discrimination and its effects within most stigmatized and underrepresented LGBTQI+ groups.

As a second and related limitation, the sample size was small, which may have increased the type 1 error. Results may differ for other samples of different sizes or characteristics, especially in cases where results were marginally significant. Finally, the study used a cross-sectional design which prevents fully inferring that discrimination caused changes in mental health. This is especially relevant in the case of internal shame, which could be a trait and a priori risk factor. Future research should address these limitations by using larger and more diverse samples and adopting a longitudinal design, allowing for examining changes over time and establishing causal relationships.

## 5. Conclusions

The findings of this study have important implications for clinical practice with people under the LGBTQI+ umbrella, suggesting that addressing internal shame is central to effective therapeutic interventions targeted to promote positive changes in the individual's self-perception, beliefs, and coping resources. Shaming experiences can often be deeply rooted in persistent feelings of distress and suffering that may be promptly reenacted. Given that internalized shame operates primarily at an unconscious level, clinical work with LGBTQI+ people should focus on examining the underlying unconscious conflicts and dynamics that contribute to the experience of internal shame. This may promote insight into self-defeating beliefs and feelings of inadequacy that erode the ability to cope with ongoing discrimination, ultimately leading to depression and anxiety. Addressing and repairing unresolved traumatic experiences of shame in LGBTQI+ childhood can be a complex and challenging process that may be better achieved by combining psychodynamic, trauma-focused, and compassionate therapy techniques designed to help individuals process and heal from traumatic experiences. Psychoanalytic and trauma-focused approaches may allow for examining early attachment experiences, how these experiences may have ingrained automatic or unconscious feelings of unworthiness and inadequacy, and processing the emotions that arise. Compassionate-focused techniques may help regain a sense of self-compassion and appreciation, allowing for identity resilience.

Aligned with previous findings, social support was found to mitigate the impact of discrimination on internal shame, depression, and anxiety among LGBTQI+ individuals. These findings suggest that social support may be decisive in promoting resilience and improving mental health outcomes for this population. This highlights the importance of addressing social support and building supportive networks as part of a comprehensive treatment plan for LGBTQI+ individuals. It may be helpful for clinicians to work with

clients to identify and strengthen social support networks and address any barriers to accessing social support. Such interventions should target resources for building and maintaining supportive relationships and address internalized stigma or self-worth difficulties that may prevent individuals from seeking and accepting support from others.

To prevent and promote LGBTQI+ mental health, targeting intervention at macro and meso-systemic levels is crucial. Within a society hierarchically structured around patriarchal and heteronormative values and norms, LGBTQI+ discrimination can become cultural violence. This violence takes many structural and institutionalized forms, both ostensive and covert, that function insidiously and continually in most daily interactions with others, including not only parents, family, peers, and colleagues but also culture and institutions. Considering the daily and profound nature of LGBTQI+ discrimination, this experience is expected to impact mental health substantially. As with many forms of cultural violence, LGBTQI+ poses an existential threat to identity, sense of belonging, and emotional security.

Interventions that increase social support and build community connections can reduce the negative impact of discrimination on mental health among LGBTQI+ individuals (e.g., providing access to supportive resources and communities and building social networks). School policies against bullying and inclusive LGBTQI+ curricula can protect LGBTQI+ youth. Inclusive measures, such as school harassment prevention protocols and LGBTQI+ content in the curricula, can increase attachment to adults at school and a sense of security (Greytak et al. 2013). Health and social services, such as safe youth centers, can provide emotional support and tangible assistance for transgender youth (Corliss et al. 2007; Singh et al. 2013; Reck 2009). Additionally, implementing anti-bullying and LGBTQ+ inclusive policies at school (e.g., school harassment prevention protocols, LGBTQI+ content on curricula, LGBTQI+ information on campus, and teacher intervention in bullying incidents) have been shown to have a significant protective effect, fostering a stronger attachment to an adult at school, a greater sense of security and lower absenteeism (Greytak et al. 2013; McGuire et al. 2010).

Finally, it has been shown that the interaction of LGBTQI+ individuals with LGBTQI+ communities or associations mitigates the impact of stigma on depression and suicide (Kaniuka et al. 2019). Additionally, making health and social services available and suited to the LGBTQI+ population at a community level, namely in schools and neighborhoods, may facilitate tangible assistance and support. An intersectional approach that considers how gender and sexuality intersect is essential for understanding the complexities of gender-based violence (GBV) and its impact on the LGBTQI+ population. Such an approach should consider not only physical and sexual violence but also cultural and structural forms of violence that arise from heterosexist gender norms, including discrimination. This will provide a more comprehensive understanding of GBV and allow for more effective intersectional strategies to address and prevent it.

**Supplementary Materials:** The following supporting information can be downloaded at: https://www.mdpi.com/article/10.3390/socsci12080454/s1, Figure S1: Discrimination estimated marginal means for gender and sexual orientation, Figure S2: Shame estimated marginal means for gender and sexual orientation; Figure S3: Depression estimated marginal means for gender and sexual orientation, Figure S4: Anxiety estimated marginal means for gender and sexual orientation; Figure S5: Social support estimated marginal means for gender and sexual orientation.

**Author Contributions:** Conceptualization, J.C; methodology, J.C. and T.M.P.; formal analysis, J.C. and T.M.P.; writing—original draft preparation, J.C. and T.M.P.; writing—review and editing, J.C. and T.M.P. All authors have read and agreed to the published version of the manuscript.

**Funding:** This work was supported by the Portuguese Foundation for Science and Technology (FCT) in the framework of the Strategic Funding UIDB/05380/2020.

**Institutional Review Board Statement:** The study was conducted in accordance with the Declaration of Helsinki and approved by the Ethics Committee of the Lusófona University (CEDIC) (Approval Code: ATA nº 4/2019).

**Informed Consent Statement:** Informed consent was obtained from all subjects involved in the study.

**Data Availability Statement:** The data presented in this study are available on request from the corresponding author. The data are not publicly available due to ethical issues.

**Acknowledgments:** The authors would like to thank Ana Filipa Ferreira for help with the literature review and data collection.

**Conflicts of Interest:** The authors declare no conflict of interest. The funders had no role in the design of the study; in the collection, analyses, or interpretation of data; in the writing of the manuscript; or in the decision to publish the results.

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
