# Peer review of "Gender, Shame, and Social Support in LGBTQI+ Exposed to Discrimination: A Model for Understanding the Impact on Mental Health"

_socsci, doi:10.3390/socsci12080454_

Round 1
Reviewer 1 Report
Overall, the research is very well designed and it contributes to filling the gap in information on the mental health status of LGBTQ+ individuals within an intersectional perspective and accounting for gender based violence.
The research sample, while convey a strong voice ont he current realities and experiences, it falls short in representing the the potentially more vulnerable communities and who might not have access to social media. Snowball recruitment might also indicate to a certain network that is formed. The most vulnerable might gave been missed as well here.
The article acknowledges this non representation of the most marginalized especially from a race based perspective.
This might be an indicator that realities are even more serious than the results shared by the research.
I strongly recommend an amplification of this limitation and its impact on the results. The conclusion might need to better reflect on this.
Author Response
We appreciate your insightful comments and thoughtful critique of our work.
In our revised manuscript, we have expanded the discussion of the limitations of our sampling procedures as follows:
"Despite the relevant contributions to the existing literature, our study has limitations that should be mentioned. At first, it fails to fully represent the breadth and diversity of the LGBTQI+ community, as it was limited to specific sociodemographic characteristics such as income and education level. Participants are mostly white with higher education and average income. Thus, findings may not be generalized to other groups within the LGBTQI+ community unrepresented in our sample, especially those that accumulate minority and discrimination-related stressors, such as transgender and non-binary, those racialized, and with low income. Existing literature and evidence highlight increased vulnerabilities at the intersection of gender, race, and sexual minority statuses. Due to these sampling constraints, the severity of discriminatory experiences and their impact on mental health could be underestimated. Moreover, different groups within these intersections may experience unique barriers when seeking social support, specifically those with multiple marginalized identities, such as racialized transgender women. Future research must strive to encompass these diverse experiences following a comprehensive understanding of interceptional discrimination and its effects within most stigmatized and underrepresented LGBTQI+ groups.
As a second and related limitation, the sample size was small, which may have increased the type 1 error. Results may differ for other samples of different sizes or characteristics, especially in cases where results were marginally significant. Finally, the study used a cross-sectional design which prevents fully inferring that discrimination caused changes in mental health. This is especially relevant in the case of internal shame, which could be a trait and a priori risk factor. Future research should address these limitations by using larger and more diverse samples and adopting a longitudinal design, allowing for examining changes over time and establishing causal relationships.".
Reviewer 2 Report
The author examines the role of internal shame and social support in the experiences of discrimination and mental health among LGBTQ+ individuals. S/he/they also consider the interactive effects between gender and sexual orientation on exposure to discrimination, shame, social support, and mental health. Although the research questions asked are novel, there remain a considerable number of needed revisions for this article to be suitable for publication.
1) The Introduction could be re-framed. The first paragraph might instead be used to highlight the persistence of discrimination among the LGBTQ+ community, while also signally what major gaps in this research the present article will fill.
In this section of the paper, the author should further review some of the research on homonegativity and also make a clearer distinction between shame and homonegativity. For example, homonegativity also entails internalized stigma, and it is unclear how this conceptually differs from shame. Thus, the author will want to make this clearer.
In addition, some prior research does look at how discrimination differentially impacts LGBTQ men versus women, and thus this review of the literature needs to be expanded.
2) There are major methodological problems with the paper. A sample size of 114 is likely inadequate to generate reliable findings and statistical conclusions. The author should consider performing a statistical power analysis to ensure that the sample size is large enough to generate significant results. For example, there are only 18 LGBTQ+ men and 21 heterosexual men, which is concerning.
The author might also explore whether there is a higher quality available secondary data set that would allow her/him/them to answer the research questions.
Greater justification is needed for some measures. For example, is the exposure to discrimination inventory widely used in research on homophobic and/or sexist discrimination?
In addition, elaborate on why UNIANOVAS are used for the paper over other possible statistical techniques. In addition, there is some evidence that looking at marginal effects is more methodologically sound when examining interaction effects, which the authors should consider investigating.
Also, carefully think through the comparison groups. Has the author, for example, conducted any analyses to see how LGBTQ+ women compare to heterosexual women (versus just everyone else) for the outcome variables of shame, depressive symptoms, and anxiety symptoms? Consider various ways to perform the analyses for the interaction effects to see if a different set of findings might emerge, or if robustness is present.
3) The author should further explain what it might mean for heterosexual men, for example, to report discrimination. Such reports are inconsistent with wider data on systematic sexist and homophobic discrimination and thus the author needs to further theorize what is happening here. Greater clarity regarding the source of shame for heterosexual men would also be helpful. Moreover, the discussion of potential gender and sexual identity differences in accessing social support could be much more theoretically rich and further reference prior research on the topics.
4) The discussion of the findings more generally could be theoretically richer. For example, what do the findings from the article contribute to understandings of the minority stress model?
5) There are very few transgender people in the sample. Perhaps, the author could speculate in the conclusion how these processes might be similar to or depart for this population.
Author Response
We appreciate your insightful comments and thoughtful critique of our work.
We compile our revisions following your specific inputs.
1) The Introduction could be re-framed. The first paragraph might instead be used to highlight the persistence of discrimination among the LGBTQ+ community, while also signally what major gaps in this research the present article will fill.
The first two paragraphs from the introduction read as: "Both gender and sexuality are central to Gender-based violence (GBV), which disproportionately affects women and those whose gender identity or expression does not align with heterosexist norms (Haynes and DeShong 2017; West, 2013). The rights of people throughout the LGBTQI+ umbrella are frequently violated in many societies worldwide, exposing this population to daily experiences of discrimination and inequality (FRA 2020; Hubbard 2020; Walters et al. 2020). For example, in Portugal, despite the growing acceptance of non-heterosexuality and recent political and legislative changes, LGBTQI+ people continue to face various forms of interpersonal and institutional discrimination. LGBTQI+ discrimination can be experienced in several areas: school, social relationships, workplace, and health services (ILGA 2020; Gato et al. 2020). Evidence shows that most LGBTQI+ youth report experiences of verbal harassment at school, and some describe being physically assaulted (e.g., punched, kicked, or wounded with a weapon (Pizmony-Levy and Kosciw 2016). In addition, many LGBTQI+ youths experience rejection from parents, friends, and peers (Pew Research Center 2013). Moreover, several studies reveal that LGBTQI+ people are often the target of professional discrimination and unequal treatment in the recruitment and selection process (Ozeren 2014; Everly 2016). For example, studies found that women with LGBTQI+ identification on their CV were discriminated against, receiving 30% fewer return calls than other women (Luiggi-Hernández et al. 2015) and that LGBTQI+ people report being the target of "jokes" and sexual harassment during recruitment and selection (Mishel 2016). LGBTQI+ individuals are also exposed to prejudice in health services, which can decrease seeking help and treatment adherence, impacting their health (Dahlhamer 2016).
While most research focuses on gender-based violence (GBV) and discrimination against women and the LGBTQI+ population, there is a lack of studies exploring the phenomena from an intersectional perspective. Such perspective is crucial in understanding how gender and sexual orientation intersect to create specific vulnerabilities or protective conditions. This is often overlooked when assessing mental health impacts, resilience, and access to protective factors. This omission obscures our understanding of the unique intersections between gender and sexuality and their role in susceptibility to discrimination. It further obscures how the mental health of distinct groups is affected, the mechanisms involved, and how they can access specific resilience resources. Our study aims to bridge this gap by investigating the differential experiences of groups with various intersections of gender and sexual orientation, focusing on exposure to discrimination, internalized shame, depression, anxiety, and the protective effects of social support. Specifically, we seek to understand how GBV and other forms of targeted violence impact LGBTQI+ persons and how to provide culturally competent resources for help and safety for all survivors."
In this section of the paper, the author should further review some of the research on homonegativity and also make a clearer distinction between shame and homonegativity. For example, homonegativity also entails internalized stigma, and it is unclear how this conceptually differs from shame. Thus, the author will want to make this clearer.
Additional paragraphs in the introduction include:
"Internalized homophobia dynamics may coexist with more pervasive feelings of internal shame, amplifying feelings of inadequacy that extend beyond sexual orientation to the essence of self-value. Internal shame represents a deep-seated, crippling self-perception of unworthiness, often emerging from early interactions with caregivers and not confined solely to sexual stigma or shaming (Gilbert 2022). As a self-conscious emotion, shame is embedded in emotional socialization and strongly influenced by these primary relationships. Caregivers' socialization of shame may, to various degrees, reflect prevailing social and cultural norms, with expected variations on how social stigma is early imprinted (Tangney, Stuewig, and Mashek 2007). These foundational experiences build the template for subsequent shame experiences, calibrating the sources and triggers of shame in response to external cues. Simultaneously, these early caregiving experiences structure emotion regulation strategies, including those required to manage feelings of shame (Gross 2015). Given the profound and overarching impact such experiences have on personality and identity development, they also establish a varying degree of resilience or vulnerability to discrimination and prejudice. Hence, understanding the intricate intersection of internal shame and homonegativity can shed light on the complex interplay of risk and protective factors when confronting discrimination, including sexual and gender prejudice, and managing its repercussions on mental health (Luthar, Crossman, and Small 2015).
While evidence suggests that LGBTQI+ people are significantly impacted by the discrimination they experience, it also indicates that some protective factors (individual, relational, and community-based) can help to buffer the physical and mental health consequences of these experiences and promote their well-being (Johns et al. 2018). Among the individual protective factors is identity resilience, a stable self-schema that combines a positive appraisal of self, a sense of cohesion, continuity, and self-efficacy. Identity resilience, as identity itself, depends on the interplay between personal and social representations and experiences (Breakwell, 2020). It includes a subjective and internal representation of self-worth and value, the willingness and ability to maintain identity despite changes, and a positive self-construal of the self as distinctive from others. Consequently, identity resilience might be established on low levels of internal shame, offering a shield against the negative impacts of sexual and gender discrimination and internalized homophobia. In a study with gay men, identity resilience was negatively associated with internalized homonegativity (Breakwell and Jaspal 2022). This same study also shows that identity resilience and internalized homonegativity are negatively impacted by perceived social discrimination and positively by social support. Furthermore, evidence on the potential mechanisms underlying the relationship between discrimination and mental health among lesbians and gay men points to a trend of gender variabilities in men and women’s internalized homonegativity and rejection, which can contribute to the observed disparities (Feinstein, Goldfried, and Davila 2012)." (...) "Taken together, findings suggest that an experience of internalized shame may explain the impact of discrimination on LGBTQI+ mental health. In addition, social support may have a protective effect, buffering the impact of exposure to discrimination on mental health. Few studies explored the association between an internalized experience of shame and mental health in LGBTQI+ people (e.g., Matos et al. 2017) since most studies have focused on self-esteem and internalized homonegativity. While self-esteem and internalized homonegativity may function at a more readily and conscious level, internal shame resides at a more core, transversal, and less consciously mentalized level and, hence, may have an enduring impact on the affect directed to the self that bypasses more conscious and rational appreciations of the right to outness, to a non-confirmative identity and of personal value. Moreover, internalized homophobia and internal shame may coexist, intensifying each other. Internal shame may amplify the impact of discrimination and potentiate stigma internalization. Internalized homophobia could, in turn, foster a more profound, generalized sense of unworthiness and shamefulness. These confluence and synergetic dynamics may intensify the deleterious effects of discrimination on mental health."
In addition, some prior research does look at how discrimination differentially impacts LGBTQ men versus women, and thus this review of the literature needs to be expanded.
Also in introduction: "
Feminist approaches to GBV have examined how relations of power account for women's increased vulnerability. However, many reproduce heteronormativity, focusing primarily on intimate partner violence or sexual assault and overlooking trans, queer, and non-binary victims, failing to account for the multiple forms in which gender and sexuality are implicated in GBV (Haynes and DeShong 2017). A still sparse but increasing body of research has, however, started to explore variations in the experiences of discrimination and violence among LGBTQI+ subgroups, including those attributed to gender differences. Some Evidence points to pervasive heterosexist, gendered, and essentialist motifs and aggressions. Results from a systematic review (Rothman et al. 2011) reveal a higher prevalence of childhood and adult sexual assault victimization for lesbian or bisexual women, while men reported higher hate crime-related sexual assault. Also, even with inconsistencies, research points to variations in exposure to discrimination among monosexual and bisexual LGBTQI+ individuals (e.g., Bostwick et al. 2014). Evidence also reveals that transgender women, particularly those of color, experience even more disproportionately high levels of discrimination, suggesting an intersection between racist, heterosexist, and transphobic forms of oppression (e.g., Smart et al. 2022).
The association between LGBTQI+ gender and sexual identities and mental health outcomes has been supported extensively and consistently (Sandfort et al. 2014; Williams et al. 2021). As with exposure to GVB and discrimination, common clinical mental health symptoms among the LGBTQI+ communities seem to vary depending on gender and sexual identity. Research on the impact of discrimination on mental health in LGBTQI+ communities demonstrates that LGBTQI+ individuals have higher rates of chronic illnesses, clinical mental health symptoms, namely depression and anxiety (Han et al. 2020; Lozano-Verduzco et al. 2017), higher rates of suicide (Fontanella et al. 2015), risky sexual behaviors (Ballard et al. 2017), and substance abuse (Kelly et al. 2015). Moreover, despite inconsistencies, evidence suggests mental health outcomes might vary for sexual minorities (Bostwick et al. 2014). Adding to this, the prevalence of mental health disorders among LGBTQI+ communities also seems to change as a function of access to social support (Henry et al. 2021; Watson, Grossman, and Russell 2019) and living in a rural or urban environment (Ballard et al. 2017). Again, such disparities underscore that discriminatory experiences and impacts demand a comprehensive and intersectional inspection that considers sexual orientation and gender, race, and class and access to resilience recourses."
2) There are major methodological problems with the paper. A sample size of 114 is likely inadequate to generate reliable findings and statistical conclusions. The author should consider performing a statistical power analysis to ensure that the sample size is large enough to generate significant results. For example, there are only 18 LGBTQ+ men and 21 heterosexual men, which is concerning.
À priori power calculations were performed following recommendations (Faul et al. 2009; Faul et al. 2007), revealing the sample size is adequate to conduct the moderated mediation models (f2 = 0.25, p < 0.05, N = 104, number of predictors = 9; power = 0.95).
The author might also explore whether there is a higher quality available secondary data set that would allow her/him/them to answer the research questions.
While we appreciate and understand the importance of the suggestion, we regrettably have to postpone implementing it due to current time constraints.
Greater justification is needed for some measures. For example, is the exposure to discrimination inventory widely used in research on homophobic and/or sexist discrimination?
An additional explanation was included in the "Participants and Procedures" section, as follows: "Discrimination was assessed using a measure encompassing various discriminatory experiences beyond those solely attributed to gender and sexuality. This broader approach facilitated the assessment of intersectionality in discrimination experiences by revealing the additional discriminatory burden endured by LGBTQI+ individuals beyond what is experienced due to other conditions (e.g., financial status, appearance). This measurement strategy also facilitates the soundness and spectrum of the comparative analysis of discrimination between cis-heterosexual and LGBTQI+ individuals and between men and women, offering a more nuanced understanding of the discriminatory phenomena across different identities. Given the study's aims and considering that we intended to explore if the LGBTQI+ and gender (being women) conditions resulted in an increased vulnerability to discrimination, only participants (heterosexual or LGBTQI+) reporting at least one experience of discrimination were included in the study."
In addition, elaborate on why UNIANOVAS are used for the paper over other possible statistical techniques. In addition, there is some evidence that looking at marginal effects is more methodologically sound when examining interaction effects, which the authors should consider investigating.
2-way and UNIANOVAs were used. We also analyzed estimated marginal means plots for possible interaction effects. These effects were further explored using moderated mediation analyses using Process. Revised results read as follows: "Results from a two-way ANOVA revealed significant univariate effects of sexual orientation on discrimination, F(3,103) = 5.03, p < .05, η2 = 0.05, and shame, F(3,103) = 34.68, p < .001, η2 = 0.25. LGBTQI+ participants reported higher levels of discrimination and shame than heterosexual participants (see Table 2). No univariate effects were found on depressive and anxiety symptoms and social support. No univariate effects of gender were found on discrimination, depressive and anxiety symptoms, shame, and social support (see Table 2). Significant effects were found on shame for the interaction between gender and sexual orientation (see Table 3). The interaction effects were inspected with a UNINOVA using an interaction variable composed of gender x sexual orientation."
Also, carefully think through the comparison groups. Has the author, for example, conducted any analyses to see how LGBTQ+ women compare to heterosexual women (versus just everyone else) for the outcome variables of shame, depressive symptoms, and anxiety symptoms? Consider various ways to perform the analyses for the interaction effects to see if a different set of findings might emerge, or if robustness is present.
Results from ANOVAS allow for these comparisons.
3) The author should further explain what it might mean for heterosexual men, for example, to report discrimination. Such reports are inconsistent with wider data on systematic sexist and homophobic discrimination and thus the author needs to further theorize what is happening here. Greater clarity regarding the source of shame for heterosexual men would also be helpful. Moreover, the discussion of potential gender and sexual identity differences in accessing social support could be much more theoretically rich and further reference prior research on the topics.
Additional information on the literature review relevant to these topics was included: "Despite heterosexual men’s privileged social and cultural position, their vulnerability to discrimination experiences is not incompatible with broader data on systemic sexist and homophobic discrimination. Our findings may be better understood given the highly subjective character of the discrimination and shaming experiences. Hegemonic masculinity expectations and pressures (Connell and Messerschmidt 2005) may contribute to exposure or vulnerability to discrimination for heterosexual men who deviate from these norms. Research demonstrates that men's identity concerns often revolve around the threat of violating traditional masculine roles, which can elicit intense feelings of anxiety, shame, and humiliation (Vandello and Bosson 2013). Moreover, male gender role socialization promotes a "shame-phobic" male experience (Reilly et al. 2014), with consequences to mental health, namely internalized shame and depression (Rice et al. 2022), highlighting the complex relationship between shame and adherence to patriarchic masculine norms. Furthermore, some heterosexual men might perceive equality demands and achievements as threatening their privilege or status (Norton and Sommers' 2011)."
"Our findings highlight the role of internal shame, mostly overlooked until now, in the minority stress model (Meyer 2013), pointing to its critical role in mental health outcomes when facing discrimination. Furthermore, these findings underline the importance of intersectionality, as they show distinct effects of discrimination, internal shame, and social support across intersections of gender and sexual identity. Our study, hence, adds to the knowledge regarding the minority stress model (Meyer 2013) by revealing internalized shame and social support as a risk and a protective factor, respectively. Furthermore, our results suggest an interaction between these factors, showing that while internalized shame may intensify the impact of discrimination on mental health, social support may buffer this impact by decreasing the effect of internal shame on depression and anxiety symptoms. In sum, results call for an expanded and nuanced understanding of minority stress that includes both conscious and unconscious experiences of internalized negativity, particularly shame, and the role of social support in buffering these experiences from an intersectional perspective. Differences related to gender and sexual identity should be further explored in future research."
4) The discussion of the findings more generally could be theoretically richer. For example, what do the findings from the article contribute to understandings of the minority stress model?
A revised version of this discussion reads as follows: "Our findings highlight the role of internal shame, mostly overlooked until now, in the minority stress model (Meyer 2013), pointing to its critical role in mental health outcomes when facing discrimination. Furthermore, these findings underline the importance of intersectionality, as they show distinct effects of discrimination, internal shame, and social support across intersections of gender and sexual identity. Our study, hence, adds to the knowledge regarding the minority stress model (Meyer 2013) by revealing internalized shame and social support as a risk and a protective factor, respectively. Furthermore, our results suggest an interaction between these factors, showing that while internalized shame may intensify the impact of discrimination on mental health, social support may buffer this impact by decreasing the effect of internal shame on depression and anxiety symptoms. In sum, results call for an expanded and nuanced understanding of minority stress that includes both conscious and unconscious experiences of internalized negativity, particularly shame, and the role of social support in buffering these experiences from an intersectional perspective. Differences related to gender and sexual identity should be further explored in future research."
5) There are very few transgender people in the sample. Perhaps, the author could speculate in the conclusion how these processes might be similar to or depart for this population.
A revised version of this discussion reads as follows: "Despite the relevant contributions to the existing literature, our study has limitations that should be mentioned. At first, it fails to fully represent the breadth and diversity of the LGBTQI+ community, as it was limited to specific sociodemographic characteristics such as income and education level. Participants are mostly white with higher education and average income. Thus, findings may not be generalized to other groups within the LGBTQI+ community unrepresented in our sample, especially those that accumulate minority and discrimination-related stressors, such as transgender and non-binary, those racialized, and with low income. Existing literature and evidence highlight increased vulnerabilities at the intersection of gender, race, and sexual minority statuses. Due to these sampling constraints, the severity of discriminatory experiences and their impact on mental health could be underestimated. Moreover, different groups within these intersections may experience unique barriers when seeking social support, specifically those with multiple marginalized identities, such as racialized transgender women. Future research must strive to encompass these diverse experiences following a comprehensive understanding of interceptional discrimination and its effects within most stigmatized and underrepresented LGBTQI+ groups.
As a second and related limitation, the sample size was small, which may have increased the type 1 error. Results may differ for other samples of different sizes or characteristics, especially in cases where results were marginally significant. Finally, the study used a cross-sectional design which prevents fully inferring that discrimination caused changes in mental health. This is especially relevant in the case of internal shame, which could be a trait and a priori risk factor. Future research should address these limitations by using larger and more diverse samples and adopting a longitudinal design, allowing for examining changes over time and establishing causal relationships."
Once again, we sincerely appreciate your thoughtful comments and the opportunity they present for enhancing the quality of our work. Your insights have been noted for consideration in future research and improvements.
Our best regards.